# Optimal Extraction Study of Gastrodin-Type Components from *Gastrodia Elata* Tubers by Response Surface Design with Integrated Phytochemical and Bioactivity Evaluation

**DOI:** 10.3390/molecules24030547

**Published:** 2019-02-02

**Authors:** Minhui Hu, Hui Yan, Yuanyuan Fu, Yulan Jiang, Weifeng Yao, Sheng Yu, Li Zhang, Qinan Wu, Anwei Ding, Mingqiu Shan

**Affiliations:** Jiangsu Collaborative Innovation Center of Chinese Medicinal Resources Industrialization, National and Local Collaborative Engineering Center of Chinese Medicinal Resources Industrialization and Formulae Innovative Medicine, Nanjing University of Chinese Medicine, Nanjing 210023, China; huminhui1353@163.com (M.H.); glory-yan@163.com (H.Y.); 18251878790@163.com (Y.F.); jiangyulan226@126.com (Y.J.); njweifengyao@163.com (W.Y.); yusheng1219@163.com (S.Y.); zhangliguanxiong@163.com (L.Z.); qnwyjs@163.com (Q.W.); awding105@163.com (A.D.)

**Keywords:** *Gastrodia elata* tuber (GET), Response surface methodology, Antioxidation, HUVEC, Gastrodin-type components

## Abstract

*Gastrodia elata* tuber (GET) is a popular traditional Chinese medicines (TCMs). In this study, response surface methodology (RSM) with a Box–Behnken design (BBD) was performed to optimize the extraction parameters of gastrodin-type components (gastrodin, gastrodigenin, parishin A, parishin B, parishin C and parishin E). Different from the conventional studies that merely focused on the contents of phytochemical, we gave consideration to both quantitative analysis of the above six components by HPLC and representative bioactivities of GET, including antioxidation and protection of human umbilical vein endothelial cells (HUVEC). Four independent variables (ethanol concentration, liquid-material ratio, soaking time and extraction time) were investigated with the integrated evaluation index of phytochemical contents. With the validation experiments, the optimal extraction parameters were as follows: ethanol concentration of 41%, liquid–solid ratio of 28.58 mL/g, soaking time of 23.91 h and extraction time of 46.60 min. Under the optimum conditions, the actual standardized comprehensive score was 1.8134 ± 0.0110, which was in accordance with the predicted score of 1.8100. This firstly established method was proved to be feasible and reliable to optimize the extraction parameters of the bioactive components from GET. Furthermore, it provides some reference for the quality control and extraction optimization of TCMs.

## 1. Introduction

*Gastrodia elata* Bl. is a perennial parasitic herb belonging to the Orchidaceae family. The plant is commonly found in the mountainous areas of Eastern Asia, including China, Korea and Japan [1]. *Gastrodia elata* tuber (GET), also called “Tianma” in China, was firstly recorded as a premium traditional Chinese medicine (TCM) in “Shen Nong Ben Cao Jing.” As a tonifying herbal medicine, GET has been used to extinguish wind to arrest convulsions, pacify and repress the liver yang, dispel wind and unblock the collaterals in clinic for thousands of years [2]. In modern pharmacological studies, GET has antioxidant, anti-angiogenic, neuroprotective, antidepressant, anxiolytic and sedative activities [3,4,5,6,7,8,9]. GET is also a quite favorite food that could improve body function and enhance immunity which frequently appears in the soup or porridge with chicken, pigeon, duck, fish head and so forth, in East Asia. Because of its high edible and health value, this herbal medicine is listed as one of the functional foods approved by the Ministry of Health in China [10,11].

For gastrodin (GD) is one of the predominant bioactive compounds in GET and gastrodigenin (GG, 4-hydroxybenzyl alcohol) due to its aglycone, which can exert specific pharmacological activities [12,13,14,15], it is listed as one of the quality evaluation indicators of GET in the Chinese Pharmacopoeia. Therefore, most studies with respect to the processing and extraction of GET only focused on these two chemical evaluation indexes [16,17,18,19]. However, there are some other ester derivatives of GD(s) in GET when combining with a citric acid, such as parishin A (PA), parishin B (PB), parishin C (PC) and parishin E (PE). The contents of these partial compounds were even higher than that of GD [20,21]. They also exhibited some promising pharmacological activities, including long-term potentiation protective, antiaging and antipsychotic effects [22,23,24,25], which were similar to that of GD, GG and GET. The cases might be explained by some researches that gastrodin and gastrodigenin are the major metabolites of the above parishin derivatives [26,27]. In this paper, considering their similar chemical structures (see Figure 1), these six components are grouped into gastrodin-type components, which were recognized as the material basis of GET.

In addition, as we know, the curative effect is the straightforward index for quality evaluation of herbal medicine. GET was reported as one of the earliest and most fundamental medicinal foods to treat vascular diseases including hypertension and atherosclerosis. The protection of this phytomedicine on human umbilical vein endothelial cells (HUVEC) involved in these bioactivities [28,29]. On the other hand, with many polyphenols including the gastrodin-type components mentioned above, GET exerted predominant antioxidant activity, which also played a crucial role in its efficacies and was related to many pharmacological bioactivities, such as protection on liver and kidney jury, antiaging, reducing cerebral ischemia/reperfusion injury, antihypertension, neuroprotection [30,31,32,33,34,35,36]. Hence, HUVEC protective and antioxidant properties could also be utilized to reflect GET quality.

With multiple quadratic regression equation, response surface methodology (RSM) is a novel statistical approach to solve multi-variable problems. RSM has been widely used to find out the optimum parameters of different kinds of process in TCM, such as extraction, processing and purification [37,38,39,40,41,42]. However, in almost all the relative studies, the contents of major compounds have been paid much attention, neglecting other critical bioactivities. In this study, taking both phytochemical yields and bioactivities as integrated evaluation indicators, we aim to optimize the extraction parameters of the gastrodin-type components from GET and to provide some reference for natural products extraction from other TCMs.

## 2. Results

### 2.1. Method Validation

The linearity, regression equation and linear ranges of six gastrodin-type components were investigated with standard solutions of different concentrations. The results in Table 1 showed a good linearity between the measured concentrations and their peak areas of each analyte in the linear range (R > 0.9992). Limit of Detection (LOD) and Limit of Quantity (LOQ) values for the analytes were also listed in Table 1.

Precision, repeatability and stability were evaluated by RSD values in Table 2. The RSD of the precision values of six gastrodin-type components were less than 3.27%. The extraction recovery rates of the analytes ranged from 96.99% to 101.72%, with the RSD values lower than 2.32%. RSD values for the stability and the repeatability were less than 2.45% and 2.75%, respectively. All results indicated that the developed method is stable, accurate and repeatable. This established HPLC method could be applied to simultaneous determination of GD, GG, PA, PB, PC and PE in GET samples. Representative HPLC chromatograms of standard solution and sample solution were shown in Figure 2.

### 2.2. Optimization of Extraction Procedure

The single-factor study was conducted to evaluate the effect of each parameter on the extraction rate and to determine the level range for BBD. With normalized contents of the target components, the effects of the four parameters on the yields of six phytochemicals in GET (*Y*_1_) were shown in Figure 3. From the figure, the three levels of each parameter were selected: 20%, 40%, 60% for ethanol concentration; 16, 28, 40 for liquid-solid ratio; 30 min, 45 min, 60 min for extraction time; 12 h, 24 h, 36 h for soaking time.

### 2.3. Optimization of DPPH Assay

In DPPH free radical scavenging assay, the reaction time and the concentration of DPPH solution were the fundamental parameters. The single-factor study was performed to assess the effect of these two factors on DPPH scavenging (see Figure 4). In terms of DPPH free radical scavenging rate, 50% was more rational and acceptable in condition optimization. Based on this point, the optimal DPPH concentration and reaction time were 75 μg/mL and 40 min, respectively.

### 2.4. Optimization of the Procedure

#### 2.4.1. The Statistical Analysis and Model Fitting

BBD experiment with four factors and three levels was carried out to optimize the four independent factors on the extraction of GET. Table 3 showed the design and the results of the dependent variables while Table 4 listed the comprehensive scores of all 29 runs. By the multivariate regression analysis, the fitted full quadratic models given by equation were obtained for the comprehensive evaluation as follows:*Y* = 1.80 + 0.016*X*_1_ + 0.13*X*_2_ + 0.094*X*_3_ − 0.021*X*_4_ − 0.11*X*_1_*X*_2_ + 0.13*X*_1_*X*_3_ + 0.10*X*_1_*X*_4_ − 0.20*X*_2_*X*_3_ + 0.25*X*_2_*X*_4_ + 0.042*X*_3_*X*_4_ − 0.47*X*_1_^2^ − 1.11*X*_2_^2^ − 0.42*X*_3_^2^ − 0.18*X*_4_^2^

By ANOVA and regression analysis (R2) method to predict the reliability of the model, the analysis of variance, goodness-of-fit and the adequacy of the regression model were listed in Table 5. The model p-value was less than 0.0001, indicating that the model was highly significant, acceptable and suitable for this experiment. In this model, the coefficient, the adjusted coefficient and the predicted coefficient were 0.9463, 0.8925 and 0.7511, respectively, indicating that 94.63% of the response value changes could be explained by the model. These results confirmed that the model could be adequate to explain the relationship between the response and the independent variables. The adequacy of the fit was evaluated by the lack of fit. The *F*-value of 1.09 and *p*-value of 0.5101 manifested that the lack of fit was not significant to the pure error and confirmed the suitability of regression model. Additionally, adequate precision in this study was 13.777, by calculating the ratio of the predicted values range at the design points to the average prediction error. The results proved the adequate model was discriminated.

As shown in Table 5, *p*-value of each model term indicated that the quadratic terms of *X*_1_^2^, *X*_2_^2^ and *X*_3_^2^ were found to have the largest effects on the model (*p* < 0.001). The quadratic terms of *X*_4_^2^ and *X*_2_*X*_4_ were also significant terms (*p* < 0.05) with the linear term of *X*_2_. However, the other terms’ effects were not significant (*p* > 0.05).

#### 2.4.2. Analysis of the Response Surface

The three-dimension response surface plots drawn by BBD were shown in Figure 5, which described the regression equation through a clear and intuitive approach and revealed the mutual effects of parameters on the comprehensive score and their reciprocal interactions. In each figure, the simultaneous effects of two factors on the response were shown by the plots and the other two factors were maintained at zero level.

Figure 5A represented the effects of liquid-solid ratio and ethanol concentration on the *Y* value. Both liquid-solid ratio and ethanol concentration had the positive effects on *Y*. It can be seen that *Y* mainly replied upon liquid-solid ratio that led to a curvilinear increase until 29 mL/g, then decreased. *Y* increased slightly when the ethanol concentration was from 20% to 41% and the concentration curve indicated that 41% was able to achieve the greatest increase. Figure 5B described the interaction effect of ethanol concentration and extraction time on *Y*. The effect of extraction time was essentially equivalent to the ethanol concentration. As the ethanol concentration was close to 41%, *Y* value increased greatly while the range of extraction time was from 30 min to 47 min. It can be seen in Figure 5C, the soaking time exhibited the weaker effect while ethanol concentration expressed a great effect on *Y*. As the soaking time prolonged from 0 to 24 h, the *Y* value increased. Therefore, *Y* was required to achieve a large value with the increasing of ethanol concentration, especially when the soaking time was close to 24 hours. The effects of extraction time and liquid-solid ratio on the yield of *Y* were shown in Figure 5D. It was obvious that the shorter extraction time resulted in lower *Y* value. While the liquid-solid ratio reached to a middle level, the *Y* value increased with a rise of extraction time. According to Figure 5E, it can be seen that the interaction between soaking time and liquid-solid ratio had significant effect on the *Y* value. It showed that the result was basically consistent with the preliminary experimental result and could prove the accurate value of the parameter. The effects of extraction time and soaking time on the *Y* values could be seen in Figure 5F. Apparently, the shorter the soaking time presented the lower the *Y* value. In general, the soaking time had weak effect on the *Y* value, otherwise the liquid-solid ratio had the most significant effect.

#### 2.4.3. Verification of Predictive Model

By Design-Expert 8.0 software, the optimum conditions for extraction of the gastrodin-type components were obtained and presented as follows: the ethanol concentration of 41%, the liquid-solid ratio of 28.58 mL/g, the soaking time of 23.91 h and the extraction time of 46.60 min. Under the conditions, the predicted comprehensive score (*Y*) was 1.8100. To verify the availability of the model equation, triplicate confirmatory experiments were carried out, which resulted in GD content of 1.95 ± 0.01 mg/g, GG content of 0.69 ± 0.02 mg/g, PA content of 14.02 ± 0.05 mg/g, PB content of 5.00 ± 0.13 mg/g, PC content of 1.40 ± 0.03 mg/g, PE content of 4.31 ± 0.03 mg/g, DPPH IC_50_ of 24.41 ± 0.33 mg/mL and HUVEC damage repair rate of 37.27 ± 0.15%. The results of verification of predictive model were listed in Table 6. The experimental comprehensive score (*Y*) was 1.8134 ± 0.0110, with no significance to the predicted one. Therefore, the model based on integrated chemical and pharmacological evaluation was suitable for the optimization of extraction process of GET. The confirmatory experiment proved that the model was adequate for the extraction process.

## 3. Materials and Methods

### 3.1. Chemicals and Materials

GETs were obtained from Liuan City, Anhui Province China and then identified by Qinan Wu in Nanjing University of Chinese Medicine (NJUCM). The voucher specimen was stored at the herbarium of NJUCM. GD (95.4% purity) and GG (98.5% purity) were purchased from National Institute for the Control of Pharmaceutical and Biological Products. PA, PB, PC and PE were purchased from Nanjing Jinyibai Biotechnology Co., Ltd. (Nanjing, China), whose purities were more than 98% by HPLC. 1,1-diphenyl-2-picrylhydrazyl (DPPH) was purchased from TCI Chemical Industry Development (Shanghai, China). HUVEC were purchased from Shanghai Baili Biotechnology Co., Ltd. (Shanghai, China). 3-(4,5-Dimethylthiazol-2-yl)-2,5-diphenyltetrazolium bromide (MTT) was purchased from Shanghai Aladdin Biotechnology Co., Ltd. (Shanghai, China). HPLC-grade methanol was provided by Merck Serono Co., Ltd. (Darmstadt, Germany). Ultrapure water was obtained by Milli-Q super purification system (Millipore, Bedford, MA, USA). All other chemicals were of analytical grade.

### 3.2. Preparation of Standard Solutions

Mixed standard stock solution containing GD, GG, PA, PB, PC and PE was prepared in 2% methanol with the final concentrations of 20.8 μg/mL, 17.2 μg/mL, 19.5 μg/mL, 16.8 μg/mL, 20.2 μg/mL and 23.8 μg/mL, respectively. This solution was diluted with 2% methanol to appropriate concentrations for linearity test. All the standard solutions were stored in a refrigerator at 4 °C before use.

### 3.3. Sample Preparation

All the GET samples were powdered and passed through an 80 mesh screen. 0.5 g of each sample was accurately weighed and mixed with aqueous ethanol of different concentrations and different volumes at room temperature. The solutions were soaked for different time ranging from 12 to 36 hours and then extracted by ultrasonic method for 30 to 60 minutes. After extraction, each solution was centrifuged at 4000 rpm for 5 minutes and the supernatant was evaporated to dryness. Then the residual was dissolved with 2% methanol in a 10 mL volumetric flask. After centrifugation at 12,000 rpm and filtration to remove the residue by using filter paper, the sample solution was stored in a refrigerator at 4 °C. Extraction details of each sample are described in Table 7.

### 3.4. Determination of the Gastrodin-Type Components

#### 3.4.1. Chromatographic Conditions

A Waters e2695 HPLC system (Waters Corporation, Milford, MA, USA) with a Waters Symmetry C18 column (4.6 mm × 250 mm, 5 μm) was used for separation. Mobile phase was composed of 0.04 M formic acid (A) and methanol (B) with gradient elution as follows: 0–10 min, 2% B; 10–60 min, 2–40% B; 60–75 min, 40% B. The column temperature was maintained at 30 °C. The flow rate was 0.8 mL/min and the injection volume was 10 μL. The detection wavelength was set at 270 nm.

#### 3.4.2. Method Validation

The HPLC method was validated in terms of linearity, sensitivity, precision, repeatability, stability and accuracy.

The linearity of each analyte was assessed by plotting its calibration curve with different concentrations and the corresponding peak areas. The standard solution of individual analyte was diluted gradually to determine its LOD and LOQ with signal-to-noise ratio of 3:1 and 10:1, respectively.

A GET sample (No. 5) was used to prepare the sample solution for the precision test, in which the solution was analyzed for six times in a day to evaluate the intra-day precision and analyzed on three consecutive days to evaluate the inter-day precision. In addition, the same sample solution was tested at 0, 1, 2, 4, 6, 9, 12, 18, 24 h to investigate the stability. In the repeatability test, six GET samples of No. 5 were extracted and analyzed according to the sample preparation procedure and the HPLC method. In these investigations, the relative standard deviation (RSD) was used to evaluate the variations. In the accuracy test, the certain amounts of the six analytes’ standards were added to the GET samples (No. 5) with six replicates. Then, these six mixed samples were treated as the above mentioned method. Recovery rate was used as the evaluation index and calculated as follows.

Recovery rate (%) = (Found amount − Known amount) ×100%/Added amount

### 3.5. Detection of Antioxidant Activity

Many researches demonstrated that DPPH assay was performed to test the ability of compounds or extracts from herbal medicines as free radical scavengers or hydrogen donors and to evaluate their antioxidant capacities. The DPPH free radical scavenging rate was determined to assess the antioxidant activity. According to a slightly modified method [43], the sample was evaporated to dryness and dissolved with methanol. Reaction solution was obtained by mixing 0.75 mL of sample solution (equivalent to GET from 2.5 mg/mL to 40 mg/mL in methanol) with 0.75 mL DPPH solution (75 μg/mL in methanol). The blank reaction solution was prepared by replacing sample solution with water as the above operation. All the reaction solutions were placed in the dark at room temperature for 40 minutes and their absorbances were measured at wavelength of 517 nm with an Infinite M200 PRO spectrophotometer (Tecan Austria GmbH, Grödig, Salzburg, Austria). The scavenging percentage was calculated as the following formula:

DPPH scavenging rate (%) = (1 − A/A_0_) × 100%

In the formula, A is defined as the absorbance of the sample reaction and A_0_ as the absorbance of the blank reaction solution. To evaluate the free radical scavenging ability of each GET sample, a regression equation of sample concentration (*X*) and scavenging rate (*Y*) was used to calculate IC_50_, half maximal inhibitory concentration. The lower IC_50_ indicated the higher free radical scavenging ability.

### 3.6. Cell Culture and Viability Assay

Primary cultures of HUVEC were performed at 37 °C under 5% CO_2_ for 2–3 days in 100 cm^2^ culture dishes with supplemented Endothelial Cell Medium (ECM) containing 5% FBS, 100 U/mL penicillin and 100 U/mL streptomycin. HUVEC were detached using the trypsin-EDTA solution and then centrifuged for precipitation at 1000 rpm for 5 minutes at room temperature. Before the cell pellets were mixed in complete medium, the supernatant was removed. After the cells were counted by the cell counting plate, they were seeded at the density of 5 × 10^4^ cells/mL in a 96-well plate and grew in cell incubator of 5% CO_2_ at 37 °C To evaluate the cell damage repair ability of GET, the cells were washed with phosphate buffered saline (PBS) and the medium was replaced by ECM containing 30 μg/mL high oxidized low density lipoprotein, human (high ox-LDL) for 24 h. After that, the medium was substituted for the sample solution, which was equivalent to 40 μg/mL of GET distributed in ECM for another 24 h. Untreated cells were compared as the model comparison [44,45].

After treatment above, the cell viability was measured by the MTT method. Briefly, MTT was dissolved at a final concentration of 5mg/mL in PBS and then added in a 96-well plate located in the cell incubator. MTT was reduced to blue formazan crystals by the metabolically active cells. After 4 h, the formed crystals were dissolved in 0.15 mL of dimethyl sulfoxide (DMSO) and the absorbance was read at 490 nm. The increasing rate of HUVEC viability was calculated as follows:

Increasing rate (%) = (A/A_0_ − 1) × 100%

In the formula, A is defined as the absorbance of the GET-treated model cell and A_0_ as the absorbance of the untreated model cell. The higher increasing rate indicated the greater protective effect on HUVEC.

### 3.7. Experimental Design for RSM

RSM with the Box-Benhnken design (BBD)(3-level, 4-factor) was used to optimize the extraction parameters, which would affect active ingredient yields and bioactivity of each GET sample. In this study, experiments were designed with the four independent extraction parameters, including ethanol concentration (*X*_1_, %), liquid-solid ratio (*X*_2_, mL/g), extraction time (*X*_3_, minute) and soaking time (*X*_4_, hour). For each parameter, the levels were coded as −1, 0, 1 with details in Table 6. The dependent variables were the contents of the six gastrodin-type components (*Y*_1_), IC_50_ of DPPH scavenging rate (*Y*_2_) and increasing rate of HUVEC viability (*Y*_3_). In order to eliminate the effects of various dimensions and units, the three dependent variables should be standardized as the following formulas:

*Y_1_* = [(S_i(GD)_ − S_min(GD)_)/(S_max(GD)_ − S_min(GD)_) + (S_i(GG)_ − S_min(GG)_)/(S_max(GG)_ − S_min(GG)_) + (S_i(PA)_ − S_min(PA)_)/(S_max(PA)_ − S_min(PA)_) + (S_i(PB)_ − S_min(PB)_)/(S_max(PB)_ − S_min(PB)_) + (S_i(PC)_ − S_min(PC)_)/(S_max(PC)_ − S_min(PC)_) + (S_i(PE)_ − S_min(PE)_)/(S_max(PE)_ − S_min(PE)_)]/6

*Y_2_* = (IC_50,i_ − IC_50,min_)/(IC_50,max_ − IC_50,min_)

*Y_3_* = (S_MTT,i_ − S_MTT,min_)/(S_MTT,max_ − S_MTT,min_)

In the formulas, S_i(GD)_, S_min(GD)_ and S_min(GD)_ represented a certain content, the minimum content and the maximum content of GD in 29 samples, respectively. The other five gastrodin-type components were expressed in the same way. S_i(MTT)_, S_min(MTT)_ and S_max(MTT)_ were a certain value, the minimum value and the maximum value of cell viability increasing rate while IC_50,i_, IC_50,min_ and IC_50,max_ were a certain value, the minimum value and the maximum value of IC_50_ in 29 samples. The three dependent variables were separately calculated. IC_50_ was a “less-better” variable while the other two were “more-better” ones. So, the comprehensive dependent variable was recorded as *Y* and calculated as follows: *Y* = *Y*_1_ − *Y*_2_ + *Y*_3_.

### 3.8. Data Analysis

Design Expert software (version 5.0.8) was used to conduct the experimental design and data analysis. To predict the response variables, a model was applied as the following formula:

*Y* = b_0_ − b_1_*X*_1_ + b_2_*X*_2_ + b_3_*X*_3_ + b_4_*X*_4_ − b_1_b_2_*X*_1_*X*_2_ + b_1_b_3_*X*_1_*X*_3_ + b_1_b_4_*X*_1_*X*_4_ − b_2_b_3_*X*_2_*X*_3_ + b_2_b_4_*X*_2_*X*_4_ + b_3_b_4_*X*_3_*X*_4_ − b_1_^2^*X*_1_^2^ − b_2_^2^*X*_2_^2^ − b_3_^2^*X*_3_^2^ − b_4_^2^*X*_4_^2^

In the formula, *Y* was the predicted dependent variable; *X*_1_, *X*_2_, *X*_3_, *X*_4_ were the predicted independent variable; b_0_ was a constant that fixed the response of the experiment; b_1_, b_2_, b_3_ and b_4_ were the regression coefficients on the linear effect terms; b_1_b_2_, b_1_b_3_, b_1_b_4_, b_2_b_3_, b_2_b_4_ and b_3_b_4_ were those on the interaction effect terms; b_1_^2^, b_2_^2^, b_3_^2^ and b_4_^2^ were those on the quadratic effect terms.

To calculate the statistical significance of the model, analysis of variance (ANOVA) and response surface analysis were selected for testing.

## 4. Conclusions

The conventional extraction optimization usually paid attention to the contents of some phytochemicals in herbal medicines. However, as the most potential indicator of quality evaluation, bioactivity has not been involved. In this study, we selected anti-oxidation (IC_50_ of DPPH radical scavenging) and protection of HUVEC (damage repair rate by MTT) as the bioactivities, which were related to many pharmacological activities of GET. With the integrated bioactivity index and quantitative index, the extraction conditions were optimized and a quadratic polynomial model was obtained for the extraction of the gastrodin-type components (GD, GG, PA, PB, PC, PE) in GET by the aid of BBD and RSM. Four main factors were employed as independent variables in the model, including ethanol concentration, liquid-solid ratio, soaking time and extraction time. Validated by the confirmatory experiments, the optimum extraction conditions were listed as follows: ethanol concentration of 41%, liquid-solid ratio of 28.58 mL/g, soaking time of 23.91 h and extraction time of 46.60 min.

As we know, the bioactive components are the material basis of the efficacy of TCMs. So, in most conditions, the contents of the major components were used for quality evaluation of TCMs and employed as the important indicators for extraction optimization of TCMs. However, just the contents of chemical components could not comprehensively reflect the quality of TCMs. The efficacy itself is actually the straight-forward index for quality evaluation. In terms of operation, cost and time, in vitro bioactivity investigation is a better choice than in vivo. In this study, we integrated the phytochemical yields and in vitro bioactivities of GET for extraction optimization. The further objective was to explore a novel strategy for quality evaluation and even application to other related research aspects of other TCMs.

## Figures and Tables

**Figure 1 molecules-24-00547-f001:**
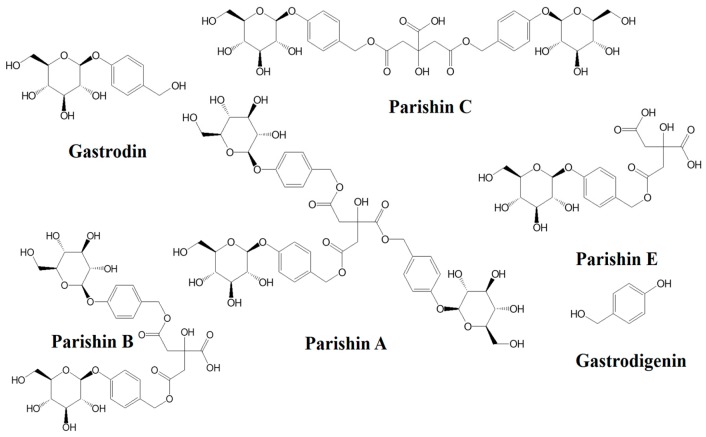
Chemical structures of the gastrodin-type components.

**Figure 2 molecules-24-00547-f002:**
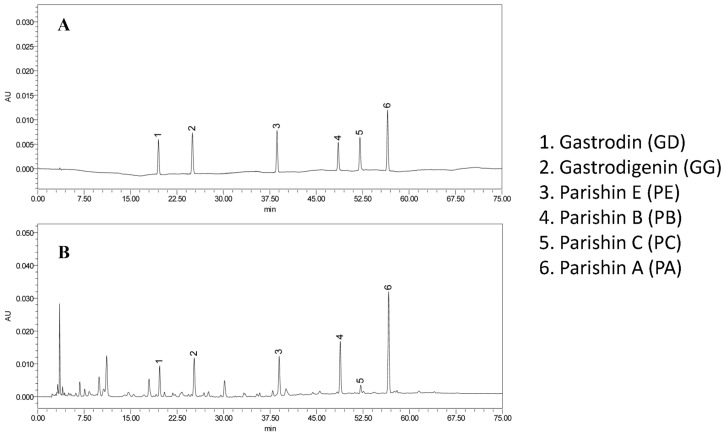
HPLC chromatograms of gastrodin-type components in (**A**) standard solution and (**B**) sample solution.

**Figure 3 molecules-24-00547-f003:**
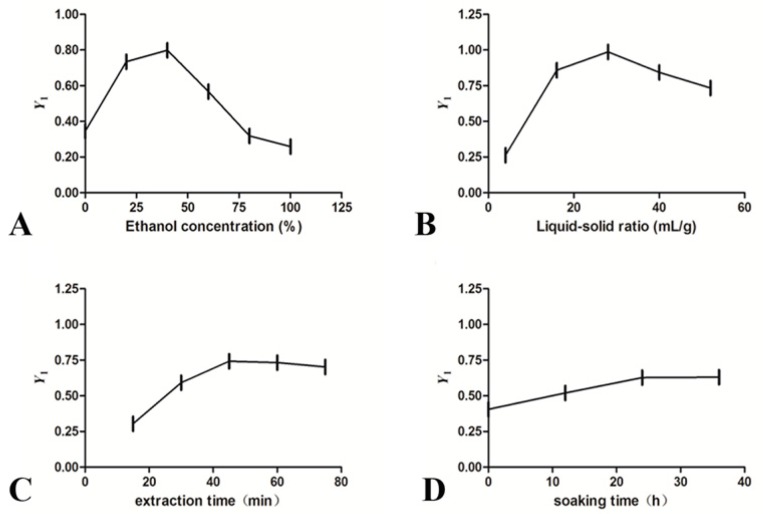
Effect of different extraction parameters (**A**: ethanol concentration, %; **B**: liquid-solid ratio, mL/g; **C**: extraction time, min; **D**: soaking time, h) on *Y*_1_.

**Figure 4 molecules-24-00547-f004:**
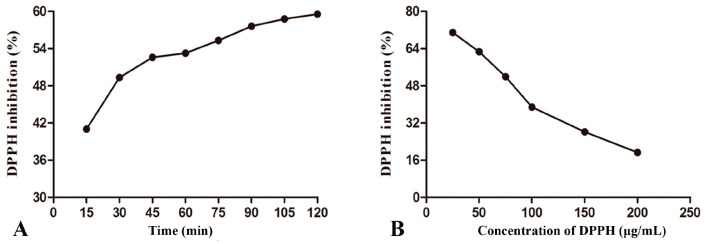
The single-factor study of DPPH assay (**A**: reaction time **B**: DPPH concentration).

**Figure 5 molecules-24-00547-f005:**
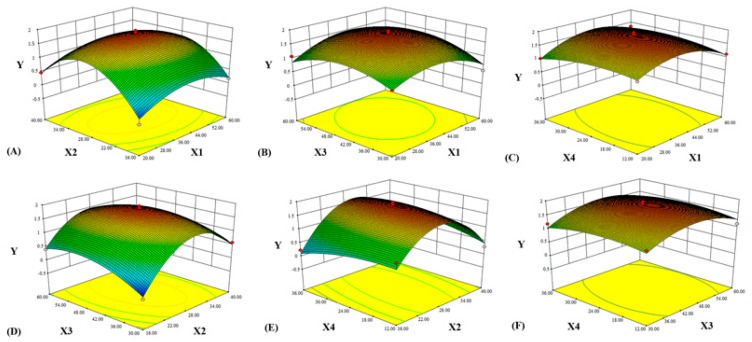
Response surface showing the interaction effects of different parameters (**A**: *X*_1_ and *X*_2_; **B**: *X*_1_ and *X*_3_; **C**: *X*_1_ and *X*_4_; **D**: *X*_2_ and *X*_3_; **E**: *X*_2_ and *X*_4_; **F**: *X*_3_ and *X*_4_) on the response *Y* (*X*_1_: ethanol concentration, %; *X*_2_: liquid-solid ratio, mL/g; *X*_3_: extraction time, min; *X*_4_: soaking time, h).

**Table 1 molecules-24-00547-t001:** The regression equations, LODs and LOQs of six gastrodin-type components.

Analyte	Regression Equation	Linear Range (μg/mL)	R^2^	LOD (μg/mL)	LOQ (μg/mL)
Gastrodin (GD)	*Y* = 1781.62*X* + 1448.30	5.21–166.64	0.9997	2.19	6.95
Gastrodigenin (GG)	*Y* = 7335.04*X* − 1097.67	2.15–68.68	0.9999	1.38	4.91
Parishin E (PE)	*Y* = 1265.20*X* − 1840.13	9.75–312.00	0.9999	0.47	1.72
Parishin B (PB)	*Y* = 1842.78*X* − 6412.23	8.42–269.28	0.9993	3.76	13.80
Parishin C (PC)	*Y* = 764.82*X* − 70.51	5.04–161.36	0.9999	2.81	6.43
Parishin A (PA)	*Y* = 956.37*X* − 6952.21	23.83–762.56	0.9997	3.92	10.60

**Table 2 molecules-24-00547-t002:** RSD of precision, stability, repeatability and accuracy for determination of six gastrodin-type components.

Analyte	Precision		Stability RSD (%)	Repeatability RSD (%)	Recovery	
	Intra-Day RSD (%)	Inter-Day RSD (%)			Mean (%)	RSD (%)
GD	2.42	2.89	2.01	2.28	98.15	1.60
GG	1.06	2.06	1.22	1.61	98.09	2.32
PE	1.22	3.27	1.03	1.21	99.72	2.14
PB	2.65	2.80	1.89	2.75	101.72	1.68
PC	2.15	1.95	2.45	2.03	96.99	0.92
PA	2.26	3.04	2.30	2.23	99.57	2.25

**Table 3 molecules-24-00547-t003:** The experimental design and results with four independent variables.

No.	*X* _1_	*X* _2_	*X* _3_	*X* _4_	GD (mg/g)	GG (mg/g)	PE (mg/g)	PB (mg/g)	PC (mg/g)	PA (mg/g)	IC_50_ (mg/mL)	S_MTT_ (%)
1	0	1	0	1	1.88	0.57	2.79	3.85	1.43	13.75	37.80	29.34
2	−1	0	1	0	1.09	0.43	2.90	2.20	0.81	7.00	26.05	38.65
3	0	0	1	1	1.74	0.63	4.66	3.65	1.39	13.59	28.11	30.87
4	0	0	−1	−1	1.91	0.61	5.05	3.98	1.47	14.32	32.68	34.22
5	−1	0	0	−1	1.64	0.65	4.30	3.27	1.20	10.79	29.92	35.75
6	−1	0	0	1	1.57	0.23	4.26	3.30	1.26	9.88	28.27	33.28
7	0	0	1	−1	1.83	0.65	4.87	3.83	1.45	14.02	28.93	28.52
8	0	0	−1	1	1.73	0.62	4.64	3.68	1.34	13.4	30.31	32.74
9	1	0	0	−1	1.73	0.63	3.17	2.84	0.80	10.71	29.61	36.81
10	0	0	0	0	1.84	0.65	4.93	3.86	1.43	14.27	22.07	37.63
11	1	−1	0	0	1.64	0.61	2.73	2.89	0.83	11.12	34.26	22.15
12	0	1	−1	0	1.85	0.65	4.95	3.96	1.45	14.29	36.24	23.58
13	1	0	−1	0	1.53	0.12	2.81	2.67	0.74	10.14	28.97	28.30
14	1	0	0	1	1.74	0.62	3.66	3.44	1.06	12.47	26.3	35.37
15	−1	1	0	0	1.60	0.58	4.18	3.16	1.17	10.57	31.76	22.81
16	0	0	0	0	1.94	0.61	5.01	3.88	1.48	14.39	21.87	38.62
17	0	−1	0	−1	1.76	0.62	4.69	3.76	1.35	13.61	34.55	29.55
18	0	−1	1	0	1.55	0.55	4.13	3.26	1.21	12.18	39.91	28.23
19	−1	0	−1	0	1.44	0.55	3.90	3.08	1.13	10.37	27.04	29.91
20	0	0	0	0	1.86	0.67	4.96	3.83	1.42	14.26	21.23	38.28
21	0	0	0	0	1.69	0.62	4.57	3.59	1.30	13.14	22.08	32.14
22	0	1	0	−1	1.62	0.49	4.24	3.45	1.26	11.85	38.77	24.55
23	0	0	0	0	1.86	0.59	4.84	3.76	1.40	13.92	21.58	36.16
24	0	−1	−1	0	1.62	0.59	4.27	3.38	1.22	12.66	44.65	18.13
25	0	−1	0	1	1.71	0.63	4.55	3.62	1.31	13.21	42.43	24.20
26	−1	−1	0	0	0.81	0.32	2.16	1.67	0.60	5.53	35.93	25.01
27	0	1	1	0	1.80	0.65	4.86	3.90	1.35	14.04	41.54	25.66
28	1	0	1	0	1.73	0.55	3.18	3.12	0.86	11.53	29.56	36.61
29	1	1	0	0	1.64	0.64	3.37	3.10	0.93	11.24	33.75	24.90

**Table 4 molecules-24-00547-t004:** Box–Behnken design for independent variables and observed responses.

No.	*Y* _1_	*Y* _2_	*Y* _3_	*Y*	No.	*Y* _1_	*Y* _2_	*Y* _3_	*Y*
1	0.7996	0.7075	0.5466	0.6387	16	0.9723	0.0273	0.9986	1.9436
2	0.2836	0.2058	1.0000	1.0778	17	0.8824	0.5687	0.5568	0.8704
3	0.8800	0.2938	0.6208	1.2070	18	0.7084	0.7976	0.4924	0.4032
4	0.9742	0.4889	0.7844	1.2696	19	0.6167	0.2481	0.5741	0.9427
5	0.7345	0.3711	0.8586	1.2220	20	0.9584	0.0000	0.9821	1.9405
6	0.5910	0.3006	0.7386	1.0290	21	0.8346	0.0363	0.6831	1.4813
7	0.9439	0.3288	0.5065	1.1216	22	0.7239	0.7489	0.3131	0.2880
8	0.8634	0.3877	0.7119	1.1876	23	0.9120	0.0149	0.8786	1.7756
9	0.5682	0.3578	0.9103	1.1207	24	0.7585	1.0000	0.0000	−0.2415
10	0.9519	0.0359	0.9505	1.8666	25	0.8448	0.9052	0.2960	0.2356
11	0.5405	0.5564	0.1959	0.1801	26	0.0606	0.6277	0.3354	−0.2317
12	0.9659	0.6409	0.2658	0.5907	27	0.9254	0.8672	0.3669	0.4251
13	0.3291	0.3305	0.4955	0.4941	28	0.5915	0.3557	0.9009	1.1368
14	0.7206	0.2165	0.8403	1.3444	29	0.6229	0.5346	0.3300	0.4183
15	0.6827	0.4496	0.2281	0.4612					

**Table 5 molecules-24-00547-t005:** ANOVA for response surface quadratic model analysis of variance.

Source	Sum of Squares	Degree of Freedom	Mean Square	*F*-Value	*p*-Value
Model	9.6300	14	0.6900	17.61	<0.0001
*X_1_*	0.0031	1	0.0031	0.08	0.7817
*X_2_*	0.2100	1	0.2100	5.5	0.0342
*X_3_*	0.1100	1	0.1100	2.72	0.1215
*X_4_*	0.0052	1	0.0052	0.13	0.7204
*X_1_X_2_*	0.0520	1	0.0520	1.32	0.2692
*X_1_X_3_*	0.0640	1	0.0640	1.65	0.2199
*X_1_X_4_*	0.0430	1	0.0430	1.11	0.3096
*X_2_X_3_*	0.1600	1	0.1600	4.2	0.0595
*X_2_X_4_*	0.2400	1	0.2400	6.22	0.0258
*X_3_X_4_*	0.0070	1	0.0070	0.18	0.6783
*X_1_^2^*	1.4200	1	1.4200	36.33	<0.0001
*X_2_^2^*	8.0300	1	8.0300	205.52	<0.0001
*X_3_^2^*	1.1200	1	1.1200	28.64	0.0001
*X_4_^2^*	0.2000	1	0.2000	5.1	0.0404
Residual	0.5500	14	0.0390		
Lack of Fit	0.4000	10	0.0400	1.09	0.5101
Pure Error	0.1500	4	0.0370		
Cor Total	10.1700	28			
R^2^	0.9463				
Adjusted R^2^	0.8925				
Predicted R^2^	0.7511				
Adequate precision	13.777				

**Table 6 molecules-24-00547-t006:** The results of verification of predictive model.

No.	GD (mg/g)	GG (mg/g)	PE (mg/g)	PB (mg/g)	PC (mg/g)	PA (mg/g)	IC_50_ (mg/mL)	S_MTT_ (%)
1	1.946	0.676	4.335	4.844	1.428	14.050	24.50	37.44
2	1.946	0.707	4.310	5.070	1.370	13.958	24.04	37.15
3	1.945	0.688	4.277	5.082	1.416	14.054	24.68	37.21

**Table 7 molecules-24-00547-t007:** Variables and experimental design levels for RSM.

Independent Variables	Coded Symbols	Levels		
		−1	0	1
Ethanol concentration (%)	*X* _1_	20	40	60
Liquid-solid ratio (mL/g)	*X* _2_	16:1	28:1	40:1
Extraction time (min)	*X* _3_	30	45	60
Soaking time (h)	*X* _4_	12	24	36

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
