# Peer review of "Optimal Extraction Study of Gastrodin-Type Components from *Gastrodia Elata* Tubers by Response Surface Design with Integrated Phytochemical and Bioactivity Evaluation"

_molecules, 2019, doi:10.3390/molecules24030547_

Round 1

Reviewer 1 Report

The work deals with the optimization of the extraction process of bioactives from tubers of Gastrodia elata, an important product of traditional Chinese Medicine.

It  is interesting and generally well done and deserves attention for a possible publication on the journal "Molecules".

Some minor clarification is needed. After the reply of the Authors, to my opinion, the work can be considered for publication.

- Line 68: please, can the Authors better explain the term "decent": not excessively high, but interesting? very remarkable? other?

- Table 1: please, add, if possible, also the full name of the assayed phytochemicals together with the respective abbreviations.

- Figure 1: very good and interesting chromatograms. I have two considerations. Please add the wavelenght of detection in the Figure caption, I think the same as described in Line 226. Moreover, I see some additional compounds, generally at lower retention times than the compounds evaluated with the standards. Their amount seem significant for some peak, in comparison with the considered compounds. Do the Authors have some idea about their identity? Is it possible to add some brief sentence on this aspect?

Line 180: at this point the parameter "ultrasonic time" has been introduced. It seems that no discussion and setting up of this parameter has not been made before, especially in the paragraph regarding the analysis of the response surface, paragraph 2.4.2 and in the parameters given in Table 6. Could it be possible that this parameter is related to the "soaking time"? Do I have well understood? If yes, I ask the Authors to uniform the name of the considered parameters. Please, clarify this aspect.

Lines 215 and 216: during the sample preparation, a centrifugation step has been performed. This means that some solubilization problem has occurred. Is it possible? No filtration step has been performed before? Please, clarify.

Lines 248-251: for the scavenging of DPPH method, this stabilized free radical is known to have some problem of water solubility. Did the Authors detect some of these problems, since, if I have well understood, the reaction system was formed by a MeOH 50% solution in water? This aspect could be not secondary, for the changes and scarce stability in Absorbance due to the low solubility of DPPH.

Author Response

The work deals with the optimization of the extraction process of bioactives from tubers of Gastrodia elata, an important product of traditional Chinese Medicine.It is interesting and generally well done and deserves attention for a possible publication on the journal "Molecules".Some minor clarification is needed. After the reply of the Authors, to my opinion, the work can be considered for publication.

Minor comments:

1. Line 68: please, can the Authors better explain the term "decent": not excessively high, but interesting? very remarkable? other?

Response: Thank you for your comment. The term "decent" has been replaced by "predominant" in line 68.

2. Table 1: please, add, if possible, also the full name of the assayed phytochemicals together with the respective abbreviations.

Response: Thank you for your advice. We have revised it according to your comment.

3. Figure 1: very good and interesting chromatograms. I have two considerations. Please add the wavelenght of detection in the Figure caption, I think the same as described in Line 226. Moreover, I see some additional compounds, generally at lower retention times than the compounds evaluated with the standards. Their amount seem significant for some peak, in comparison with the considered compounds. Do the Authors have some idea about their identity? Is it possible to add some brief sentence on this aspect?

Response: Thank you for your comment. We have added the wavelength according to your suggestion. The study mainly focused on gastrodin-type components which have similar structure with gastrodin, so the additional compounds was not explicitly associated with the phytochemical and bioactivity effects identified in this study. This suggestion is very valuable, so we will continue to explore these additional components in subsequent study.

4. Line 180: at this point the parameter "ultrasonic time" has been introduced. It seems that no discussion and setting up of this parameter has not been made before, especially in the paragraph regarding the analysis of the response surface, paragraph 2.4.2 and in the parameters given in Table 6. Could it be possible that this parameter is related to the "soaking time"? Do I have well understood? If yes, I ask the Authors to uniform the name of the considered parameters. Please, clarify this aspect.

Response: Thank you for your comment. The parameter "ultrasonic time" is not related to "soaking time". Because the extraction method was ultrasonic method in this study, “extraction time” meant “ultrasonic time”. The names of the two parameters have been unified in this article according to your suggestion.

5. Lines 215 and 216: during the sample preparation, a centrifugation step has been performed. This means that some solubilization problem has occurred. Is it possible? No filtration step has been performed before? Please, clarify.

Response: Thank you for your comment. We have filtered the extraction solution to remove the residue and then no solubilization problem occured. The sentences in the article have been corrected according to your comment.

6. Lines 248-251: for the scavenging of DPPH method, this stabilized free radical is known to have some problem of water solubility. Did the Authors detect some of these problems, since, if I have well understood, the reaction system was formed by a MeOH 50% solution in water? This aspect could be not secondary, for the changes and scarce stability in Absorbance due to the low solubility of DPPH.

Response: Thank you for your comment. We have found that stabilized free radical of DPPH have some problem of water solubility during the experiment. The samples for scavenging of DPPH assay were re-evaporated and then dissolved by methanol in the experiment, so the reaction system was formed by methanol solution. The sentences in the article have been corrected according to your comment.

Reviewer 2 Report

I don't recommend the paper for publication in the current status. Some minor issues are listed below.

1.      Article needs an minor English correction

2.      Some notes should be clear. For examples,

·  in line 52 the purpose of number 2 in text is not clear

·  in line 58 sentence : The cases might be explained by some pharmacokinetic facts that gastrodin and gastrodigenin are the major metabolites of the parishin derivatives above.

This is not a pharmacokinetic fact.

·  Figure 1 needs to be higher resolution and larger dimensions

·  in line 68 sentence : GET exerted decent antioxidant activity... Decent should be rephrased

·  in line 87 sentence : LOD and LOQ values for the analytes were also listed in Table 1. LOD and LOQ values are mentioned for the first time, needs explanation

·  Results of GD, GG, PA, PB, PC, PE content, DPPH IC50,  HUVEC damage repair rate are not listed anywhere, only mentioned in Verification of predictive model subsubdivision. What is the reason for that? Only triplicate confirmatory experiments were carried out, which is insufficient for such a title that includes Phytochemical and Bioactivity Evaluation

·  The writing seems incomplete, especially, in the Conclusion section

·  Objective conclusions based on results.

Author Response

I don't recommend the paper for publication in the current status. Some minor issues are listed below.

Minor comments:

1. Article needs an minor English correction.

Response: Thank you for your comment. This article has been checked and revised by a native speaker.

2. In line 52 the purpose of number 2 in text is not clear.

Response: Thank you for your comment. It has been corrected according to your suggestion.

3. In line 58 sentence : The cases might be explained by some pharmacokinetic facts that gastrodin and gastrodigenin are the major metabolites of the parishin derivatives above. This is not a pharmacokinetic fact.

Response: Thank you for your comment. The metabolic literature of related on GET could express that gastrodin and gastrodigenin are the major metabolites of the parishin derivatives above. It has been corrected according to your suggestion.

4. Figure 1 needs to be higher resolution and larger dimensions.

Response: Thank you for your comment. Its resolution is modified from 300 dpi to 600 dpi according to your suggestion.

5. In line 68 sentence : GET exerted decent antioxidant activity... Decent should be rephrased.

Response: Thank you for your comment. The term "decent" has been replaced by "predominant" in line 68.

6. In line 87 sentence : LOD and LOQ values for the analytes were also listed in Table 1. LOD and LOQ values are mentioned for the first time, needs explanation.

Response: Thank you for your comment. The complete explanation has been added according to your comment.

7. Results of GD, GG, PA, PB, PC, PE content, DPPH IC50, HUVEC damage repair rate are not listed anywhere, only mentioned in Verification of predictive model subsubdivision. What is the reason for that? Only triplicate confirmatory experiments were carried out, which is insufficient for such a title that includes Phytochemical and Bioactivity Evaluation.

Response: Thank you for your comment. Results of GD, GG, PA, PB, PC, PE content, DPPH IC50, HUVEC damage repair rate have been listed in Table 6 according to your suggestion. In the majority of RSM studies, triplicate confirmatory experiments were adopted [1-3]. So, we think triplicate confirmatory experiments are sufficient to prove the reliability and stability of the model.

1. Wu, J.K.; Yu, D.; Sun, H.F.; Zhang, Y.; Zhang, W.W.; Meng, F.J.; Du, X.W. Optimizing the extraction of anti-tumor alkaloids from the stem of berberis amurensis by response surface methodology. Ind. Crops Prod. 2015, 69, 68-75.

2. Belwal, T.; Dhyani, P.; Bhatt, I.D.; Rawal, R.S.; Pande, V. Optimization extraction conditions for improving phenolic content and antioxidant activity in berberis asiatica fruits using response surface methodology (rsm). Food Chem. 2016, DOI: http://dx.doi.org/10.1016/j.foodchem.2016.03.081

3. Chen, Y.; Zhang, J.; Li, Q.; Wu, J.; Sun, F.; Liu, Z.; Zhao, C.; Liang, S. Response Surface Methodology for Optimizing the Ultrasound-Assisted Extraction of Polysaccharides from Acanthopanax giraldii. Chem. Pharm. Bull. 2018, 66, 785-793.

8. The writing seems incomplete, especially, in the Conclusion section. Objective conclusions based on results.

Response: Thank you for your comment. A part of objective conclusions based on results has been included in the discussion, and the conclusion has been extended according to your suggestion.

Reviewer 3 Report

The authors in this manuscript use both phytochemical yields and  bioactivities as evaluation indicators to optimize the extraction of gastrodin-type components from GET. The whole technique will provide a very useful method/tool for the GET quality control or bioactive research.

Below are some comments

1) During the preparation of standard solutions, why all the standards were dissolved in 2% methanol? Is the gastrodigenin will dissolve in this hydrophilic solution?

2) Is the GET samples are dry? If the GET are not dry before soaked in the solution, the liquid-solid ratio will not reflect the real ratio of the GET with the solvent.

3) Have the author thought about other extraction method such as reflux?

Author Response

I don't recommend the paper for publication in the current status. Some minor issues are listed below.

Minor comments:

1. During the preparation of standard solutions, why all the standards were dissolved in 2% methanol? Is the gastrodigenin will dissolve in this hydrophilic solution?

Response: Thank you for your comment. To prevent solvent effects in the HPLC performance, 2% methanol was used to dissolve the samples, which was consistent with the initial ratio of the mobile phase in the gradient elution. As other gastrodin-type components, gastrodigenin has good solubility in hydrophilic solvents and was dissolved in 2% methanol.

2. Is the GET samples are dry? If the GET are not dry before soaked in the solution, the liquid-solid ratio will not reflect the real ratio of the GET with the solvent.

Response: Thank you for your comment. Before being powdered, the moisture content was measured to be 3.04%, far less than 15%, the upper limit of GET in the present Chinese Pharmacopoeia. In our opinion, the moisture content was too low and had little effect on the ethanol concentration in the experiment, so the liquid-solid ratio was consistent with the designed ratio.

3. Have the author thought about other extraction method such as reflux?

Response: Thank you for your comment. We have experimented and compared the content of the components by ultrasonic extraction with it by reflux extraction. In comparison, ultrasonic extraction method is convenient and efficient while saving energy and time, so we chose ultrasound extraction for research.